# Evaluation of the Acousto-Optic Figure of Merit and the Maximum Value of the Elasto-Optic Constant of Liquids

**DOI:** 10.3390/ma17122810

**Published:** 2024-06-08

**Authors:** Pavel A. Nikitin, Vitold E. Pozhar

**Affiliations:** Scientific and Technological Centre of Unique Instrumentation RAS, 117342 Moscow, Russia; nikitin.pavel.a@gmail.com

**Keywords:** acousto-optic, elasto-optic constant, liquid

## Abstract

The elasto-optic properties of liquids on the basis of the first principles of acousto-optics were theoretically investigated. A relationship for calculating the elasto-optic constant of liquids using only the refractive index was obtained. The refractive index values corresponding to the maximum elasto-optic constant for polar and nonpolar liquids were determined. Calculations for about 100 liquids were performed and compared with known experimental data. This study significantly extends our understanding of the acousto-optic effect and has practical applications for predicting the elasto-optic constant of a liquid and estimating its wavelength dispersion.

## 1. Introduction

Acousto-optic devices are used today in many fields, e.g., in laser technology, medicine and optical information-processing systems [1,2,3]. Due to the formation of a phase structure in the medium under the influence of ultrasound, acousto-optic devices allow for the parameters of the light beam to be controlled in real time. It is assumed that the ultrasound frequency is less than the inverse time of the relaxation processes [4]. For the rotation of aspherical molecules due to the ultrasound wave, the relaxation time is about 0.1–10 ns [5]. Therefore, the ultrasound frequency has to be less than 100 MHz, which is reasonably accurate for the majority of practical applications. The greatest success was achieved by using birefringent single crystals (e.g., paratellurite, TeO_2_) as the medium for the acousto-optic interaction [6]. Such crystals also have a pronounced acoustic anisotropy, which makes it possible to select the optimal propagation directions of light and ultrasound beams for a given task.

The energy efficiency of acousto-optic devices is determined by the coefficient $M_{2}$, which characterizes the acousto-optic figure of merit of the medium in which the diffraction of radiation by ultrasound takes place. This coefficient is a complex function of the components $p_{ij}$ of the elasto-optic tensor [7], the speed of sound *V*, the density of the medium ρ and the refractive index *n*. For reasons of simplicity, the effective elasto-optic constant *p* is used. In this case, the formula for the acousto-optic figure of merit reads as follows [6]:(1)$$M_{2}=\frac{p^{2}n^{6}}{\rho V^{3}}.$$

Note that all known values of the components $p_{ij}$ of the elasto-optic tensor of crystals and liquids are less than one [8]. Therefore, the value of the effective elasto-optic constant *p*, which is a combination of the $p_{ij}$ values, is also bounded from above. In our opinion, this fact is of fundamental importance and, therefore, requires a detailed investigation. In this study, the maximum value of the effective elasto-optic constant was determined using liquids as an example. This is not only important from a theoretical point of view but can also increase the energy efficiency of acousto-optic devices by choosing the optimal interaction medium.

## 2. Theory of the Elasto-Optic Effect in Liquids

### 2.1. Elasto-Optic Effect

The elasto-optic effect consists of variation in the permittivity ε of the medium caused by its deformation. Let us define an elementary volume in the medium in the form of a cube. Its compression by Δx along an edge with length $L_{x}$ parallel to the Ox axis leads to a displacement of the matter particles *u* and, as a consequence, to a change in the density Δρ and relative deformation *S*:(2)$$u=x\frac{\Delta x}{L_{x}}=x\frac{\Delta \rho }{\rho },$$
(3)$$S=\frac{du}{dx}=\frac{\Delta \rho }{\rho }.$$

At the same time, it is known that in the first approximation, the change in the dielectric impermeability ΔB (where $B=1/\varepsilon =1/n^{2}$) is proportional to the relative strain *S*, and the proportionality coefficient is the elasto-optic constant *p* [9]:(4)$$\Delta B=\frac{1}{n^{2}}-\frac{1}{{\left(n+\Delta n\right)}^{2}}\approx \frac{2\Delta n}{n^{3}}=pS.$$

It follows from (Equation 4) that the ratio of the change in the refractive index Δn to the relative strain *S* for a given medium is a constant that is independent of the strain. This constant is sometimes used and is referred to as the elasto-optic coefficient ρ∂n/∂ρ (not to be confused with the elasto-optic constant *p*) [10]:(5)$$\rho \frac{\partial n}{\partial \rho }\approx \rho \frac{\Delta n}{\Delta \rho }=\frac{\Delta n}{S}.$$

By substituting Expression (Equation 5) into (Equation 4), the relationship between the elasto-optic constant *p* and the elasto-optic coefficient ρ∂n/∂ρ can be obtained [11]:(6)$$p=\frac{2}{n^{3}}\left(\rho \frac{\partial n}{\partial \rho }\right).$$

Note, that sometimes ρ∂ε/∂ρ is used instead of ρ∂n/∂ρ, and the elasto-optic constant *p* can be calculated as follows (assuming $\varepsilon =n^{2}$):(7)$$\rho \frac{\partial \varepsilon }{\partial \rho }=2n\left(\rho \frac{\partial n}{\partial \rho }\right),$$
(8)$$p=\frac{1}{n^{4}}\left(\rho \frac{\partial \varepsilon }{\partial \rho }\right).$$

### 2.2. Review of Permittivity Models

The dependence of the permittivity ε on the density ρ is non-linear and also implicitly includes temperature and pressure. In most works, the relationships between ε and ρ, as well as the expressions for the elasto-optic coefficient ρ∂n/∂ρ, were determined. Using (Equation 8), we derive expressions for ∂ε/∂ρ and for the elasto-optic constant *p*. The models can be divided into three groups: (1) the general theories (Lorentz, Onsager, Kirkwood, Proutiere), (2) the simple approximations to the known relationships (Rocard, Looyenga) and (3) the empirical rules for the best fit of some experimental data (Gladstone–Dale, Eykman, Wahid, Meeten).

The simplest model uses the Lorentz–Lorenz Formula [12,13], which is also known as the Clausius–Mossotti relation [14,15]:(9)$$\begin{array}{cccccc} \frac{\varepsilon -1}{\varepsilon +2} & =\left(\frac{4\pi }{3}\frac{N_{A}\alpha }{M}\right)\rho \propto \rho , & \frac{\partial \varepsilon }{\partial \rho } & \propto \frac{{\left(\varepsilon +2\right)}^{2}}{3}, & p & =\frac{\left(n^{2}-1\right)\left(n^{2}+2\right)}{3n^{4}}, \end{array}$$
where *M* is the molar mass, $N_{A}$ is the Avogadro number and α is the molecular polarizability.

Let us emphasize the assumptions in the Lorentz–Lorenz model. First, it is assumed that the molecules have a spherical shape. Second, the dipole polarizability caused by the rotation of dipole molecules is neglected. And finally, the influence of neighboring molecules on each other is not taken into account.

The approximation of the Lorentz–Lorenz model was given by Rocard, who assumed (ε+2) as a constant in the Lorentz–Lorenz Formula (Equation 9) [16]:(10)$$\begin{array}{cccccc} \varepsilon -1 & \propto \rho , & \frac{\partial \varepsilon }{\partial \rho } & \propto \mathrm{const}, & p & =\frac{n^{2}-1}{n^{4}}. \end{array}$$

One more model that does not involve the shapes of the molecules was developed by Looyenga [17] and suggests the following equation as a simple approximation of the Lorentz–Lorenz relation (Equation 9) for ε≈1:(11)$$\begin{array}{cccccc} \varepsilon^{1/3}-1 & \propto \rho , & \frac{\partial \varepsilon }{\partial \rho } & \propto \varepsilon^{2/3}, & p & =\frac{3(n^{2/3}-1)}{n^{8/3}}. \end{array}$$

The Onsager formula [18,19], which is referred to as Oster’s rule in some works [20,21], is used to calculate the dielectric constant of liquids. In this model, the molecules are assumed to be a polarizable point dipole located at the center of a spherical cavity (the Onsager cavity):(12)$$\begin{array}{cccccc} \frac{\left(\varepsilon -1)(2\varepsilon +1\right)}{\varepsilon } & \propto \rho , & \frac{\partial \varepsilon }{\partial \rho } & \propto \frac{\varepsilon^{2}}{2\varepsilon^{2}+1}, & p & =\frac{\left(n^{2}-1\right)\left(2n^{2}+1\right)}{\left(2n^{4}+1\right)n^{2}}. \end{array}$$

To provide a more precise description of polar liquids, the Kirkwood model introduces a *g*-factor to consider the short-range intermolecular interaction [22,23], resulting in the following expressions [24]:(13)$$\begin{array}{cccc} \varepsilon -1 & \propto \rho (1+a\rho ), & p & =\frac{\left(n^{2}-1\right)\left(2n^{2}+1\right)}{\left(n^{2}+2\right)n^{4}}. \end{array}$$

In refining the Kirkwood model, Niedrich assumed that the molecules are not identical and that the local electric field is determined solely by the dielectric constant, independent of temperature and density [24]:(14)$$\begin{array}{cccc} \frac{\left(\varepsilon -1)(2\varepsilon +1\right)}{\varepsilon } & \propto \rho \exp(b\rho^{2}), & p & =\frac{\left(n^{2}-1\right)\left(2n^{2}+1\right)}{\left(n^{2}+2\right)n^{4}}\frac{3(n^{4}+2)}{\left(2n^{2}+1/n^{2}\right)\left(n^{2}+2\right)}. \end{array}$$

Proutiere in [25,26,27] showed that the approximation of the local electric field must be made after averaging the molecular dipole moment and that this moment is independent of density fluctuations [28]. It was established that this model gives more accurate results than others [29,30]:(15)$$\varepsilon -1\propto \rho \frac{\varepsilon \alpha }{2\bar{\varepsilon }+1-2\left(\bar{\varepsilon }-1\right)\varphi \rho \left(\alpha N_{A}/3\varepsilon_{0}M\right)},$$
(16)$$p=\frac{n^{2}-1}{n^{4}}\left[1+\frac{2{\left(n^{2}-1\right)}^{2}\varphi }{3\left(2n^{2}+1\right)-6{\left(n^{2}-1\right)}^{2}\left(\varphi -1\right)/\left(2n^{2}+1\right)}\right],$$
where $\varepsilon_{0}$ is the dielectric constant of a vacuum, the bar indicates a mean value in the bulk liquid that surrounds the Onsager cavity and $\varphi =6/\pi \sqrt{2}$ is the inverse value of a part of the available space filled by the molecular Onsager cavities.

As can be seen, numerous models describing the elasto-optic effect have been developed to date. It is worth mentioning empirical relations: Gladstone–Dale rule [21,31]:(17)$$\begin{array}{cccccc} \sqrt{\varepsilon }-1 & \propto \rho , & \frac{\partial \varepsilon }{\partial \rho } & \propto \sqrt{\varepsilon }, & p & =\frac{2(n-1)}{n^{3}}, \end{array}$$
Eykman’s rule [26], which is applicable for organic solvents within the refractive index range 1.35 < *n* < 1.5 [32]:(18)$$\begin{array}{cccccc} \frac{\varepsilon -1}{\sqrt{\varepsilon }+0.4} & \propto \rho , & \frac{\partial \varepsilon }{\partial \rho } & \propto \frac{\sqrt{\varepsilon }{\left(\sqrt{\varepsilon }+0.4\right)}^{2}}{\varepsilon +0.8\sqrt{\varepsilon }+1}, & p & =\frac{2\left(n^{2}-1\right)\left(n+0.4\right)}{n^{3}\left(n^{2}+0.8n+1\right)}, \end{array}$$
Wahid’s rule [16] for 1.3 < *n* < 1.6:(19)$$\begin{array}{cccccc} \frac{\varepsilon -1}{\varepsilon^{1/3}} & \propto \rho , & \frac{\partial \varepsilon }{\partial \rho } & \propto \frac{\varepsilon^{4/3}}{2\varepsilon +1}, & p & =\frac{3(n^{2}-1)}{n^{2}\left(2n^{2}+1\right)}. \end{array}$$
and Meeten’s rule for temperatures from −50 °C to +35 °C, for pressures from 1 to $10^{3}$ bars and over the whole visible spectrum [32]:(20)$$\begin{array}{cccc} \rho \frac{\partial \varepsilon }{\partial \rho } & =\frac{\left(\varepsilon -1)(7\varepsilon +23\right)}{30}, & p & =\frac{\left(n^{2}-1\right)\left(7n^{2}+23\right)}{30n^{4}}. \end{array}$$

Usually, the models under consideration are used to calculate the permittivity ε or molecular polarizability. Verification of the models (excluding the Proutiere model) in the experiment on compressing liquids to a pressure of 14 kbar revealed that the Lorentz–Lorenz model has the highest error rate (approximately 10%), while the Niedrich formula has the lowest (about 5%) [24,33]. It was demonstrated [29] that the Proutiere formula provides a more accurate result (with an error of about 2%) for polar liquids, such as water, compared with the Niedrich formula. At the same time, the error in Meeten’s rule is within 1% relative to experimental data for most liquids [32].

The main theoretical problem is to determine the local electric field of the molecules, taking into account the shape of the molecule and the influence of the neighboring molecules. The only way to estimate the applicable range of the model is to compare the consequences of the theory with experimental data: (1) the pressure dependence of the refractive index [24,33], (2) the light scattering in a liquid due to thermal fluctuations in the permittivity [29] and (3) the pressure dependence of the elasto-optic constant [26,32]. But even if a model gives the correct result for one physical parameter (molecular polarizability, permittivity), it does not mean that the model is correct for another physical parameter (elasto-optic constant). In fact, the elasto-optic constant comparison derived from the most known models is performed here for the first time. The results are summarized in the following section.

## 3. Comparison of Permittivity Models

The models work fine to give an accurate relationship between the permittivity ε and the density ρ of the medium. At the same time, it is interesting to compare all these models in detail with regard to the calculation of the elasto-optic constant *p*. For this purpose, the theoretical dependencies p(n), as well as experimental data from [15,26], are presented in Figure 1. It should be noted that the data for the same fluid scatter by 5–10%. Furthermore, the dynamic elasto-optic constant could even be only half as large as that determined under static conditions [34,35]. Nevertheless, as one can see, the data agree qualitatively with the models.

According to (Equation 9)–(Equation 19), the density derivative of the permittivity ∂ε/∂ρ at ε≫1 is estimated as (1) a constant (∂ε/∂ρ≈const in the Rocard and Onsager models), which is determined by the polarizability of the medium; (2) a linear dependence on the refractive index (∂ε/∂ρ∝n in the Gladstone–Dale and Eykman models); and even (3) a significantly nonlinear function ($\partial \varepsilon /\partial \rho \propto n^{4}$ in the Lorentz–Lorenz model). It should also be noted that only in the Lorentz–Lorenz (Equation 9) and Meeten (Equation 20) models, there is a plateau of p(n) at n≫2. However, there is a common feature in the considered dependencies of the elasto-optic constant *p* on the refractive index *n*. In most of the considered models, the elasto-optic constant *p* increases linearly with the refractive index *n* according to the law p∝n−1, and at n≫2, the elasto-optic constant decreases as $p\propto 1/n^{2}$. For example, in (Equation 13), (Equation 17) and (Equation 18),
(21)$$p\approx \left\{\begin{array}{cc} 2(n-1) & \mathrm{at}\;n-1\ll 1\\ 2/n^{2} & \mathrm{at}\;n\gg 2 \end{array}\right..$$

The analysis of the models allowed us to determine the value of the refractive index $n_{\mathrm{opt}}$ corresponding to the maximum value of the elasto-optic constant max(*p*). The results are summarized in Table 1.

One can see from Figure 1 that the Proutiere relation (Equation 16) describes the experimental data the most accurately, while the Lorentz–Lorenz (Equation 9) and Rocard (Equation 10) models are the crudest approximations. At the same time, the Proutiere model predicts a clearly different dependence of the elasto-optic constant p(n) on the refractive index at n>1.6. According to Formula (Equation 16), the elasto-optic constant of liquids with a high optical density is an increasing function of the refractive index, while according to other models, in contrast, it is a decreasing function. As the Lorentz–Lorenz formula was one of the first models, it gives significant errors. At the same time, the Proutiere relation is a result of the much more modern theory with an accuracy of about a few percent. Certainly, there is the significant difference between these and other models at n>1.6, but which ones are correct could not be clarified owing to a lack of experimental data.

It is important to note that the Lorentz–Lorenz model predicts the largest value of the elasto-optic constant and all experimental values of *p* are below this dependence (Equation 9). This fact was also established by Niedrich [24]. Therefore, Formula (Equation 9) can be used to estimate the maximum value of the elasto-optic constant *p*.

## 4. AO Figure of Merit of Liquids

The result obtained in the previous section is very important: the optimal value of the refractive index $n_{\mathrm{opt}}$ of the liquid at which the elasto-optic constant *p* is a maximum was determined. However, for practical applications, the value of *p* is not the only one can be used as a criterion for the selection of the optimal acousto-optic interaction medium. As follows from (Equation 1), the energy efficiency of the AO devices is determined by the AO figure of merit $M_{2}$, which is a complex parameter. Combining (Equation 1) and (Equation 6), one can obtain the following relation for $M_{2}$:(22)$$M_{2}=\frac{p^{2}n^{6}}{\rho V^{3}}=\frac{4}{\rho V^{3}}{\left(\rho \frac{\partial n}{\partial \rho }\right)}^{2}.$$

A review of the data [36] for more than 250 liquids allowed us to plot a density–velocity diagram (see Figure 2) in which the points correspond to different liquids. As can be seen, it cannot be said that a denser liquid is characterized by a greater value of the speed of sound. Therefore the further considerations in this section only apply to a single liquid. In this case, one can assume that the speed of sound is proportional to the density of the medium [37]:(23)V=w(−1+vρ),
where *w* and *v* are coefficients that are environment-specific and have the same sign (w>0, v>0 or w<0, v<0).

Under the Lorentz–Lorentz approximation, the relations (Equation 1), (Equation 9) and (Equation 23) allow one to write the expression for the acousto-optic figure of merit in the following way:(24)$$M_{2}=\frac{v}{w^{3}}\frac{1}{v\rho {\left(v\rho -1\right)}^{3}}\frac{{\left(n^{2}-1\right)}^{2}{\left(n^{2}+2\right)}^{2}}{9n^{2}}$$

Combining (Equation 24) and the relation (Equation 9) between density ρ and permittivity ε leads to the following expression for the acousto-optic figure of merit (see Figure 3):(25)$$M_{2}=\frac{v}{w^{3}}\frac{1}{v\rho^{\max}{\left[\left(v\rho^{\max}-1\right)n^{2}-\left(2+v\rho^{\max}\right)\right]}^{3}}\frac{{\left(n^{2}-1\right)}^{2}{\left(n^{2}+2\right)}^{2}}{9n^{2}},$$
(26)$$\rho^{\max}=\frac{3M}{4\pi N_{A}\alpha },$$
where $\rho^{\max}$ is the maximal value of ρ according to the Lorentz–Lorenz model (Equation 9) at infinite permittivity ε [17].

As can be seen from Figure 3, the shape of the graph $M_{2}\left(n\right)$ is determined by the sign of coefficients in the dependence V(ρ) (Equation 23). For liquids under normal conditions, as follows from [37], the free term of the linear approximation V(ρ) is negative. At the same time, the dependence V(ρ) for liquefied gases is essentially non-linear. Therefore, the slope angle and, consequently, the free term of the linear approximation depend on the operating point (temperature and pressure). Note, that infinite growth of the acousto-optic figure of merit $M_{2}$ with a decrease in the refractive index *n* is related to the extremely small sound velocity *V*, as follows from (Equation 23) and (Equation 25).

There are a few peculiarities that should be mentioned for the dependence of the normalized acousto-optic figure of merit on *n* and $v\rho^{\max}$. For $v\rho^{\max}<0$, the optimal value of the refractive index $n_{\mathrm{opt}}$ can be found as follows:(27)$$n_{\mathrm{opt}}=\sqrt{\frac{10-7v\rho^{\max}+3\sqrt{{\left(v\rho^{\max}\right)}^{2}-28v\rho^{\max}+4}}{2(-5v\rho^{\max}-4)}},$$
where the relation for the maximal value of $M_{2}$ can be easily obtained, but it is not presented because of its highly complicated nature.

It is interesting that at a high refractive index n≫1, the acousto-optic figure of merit $M_{2}\approx 1/\rho^{\max}w^{3}{\left(v\rho^{\max}-1\right)}^{3}$ seems to be independent of *n* and depends only on the density and sound velocity parameters. However, this is not the case, as $\rho^{\max}$ depends on the polarizability α, which is an optical parameter that varies with temperature and pressure. This means a limitation for the model used.

## 5. Discussion

The acousto-optic figure of merit $M_{2}$ is a complex parameter of acoustic, optic and elasto-optic properties of the medium, i.e., four items: *V*, ρ, *n* and *p*. At the same time, the models of permittivity give us simple relations p=p(n), reducing the number of parameters to three. As shown in the previous section, the structure of the relationship in (Equation 25) allows one to reduce everything to the analysis of the function of only two variables $M_{2}w^{3}/v=f\left(v\rho^{\max},n\right)$. However, it should be noted that there are forbidden zones on this dependence since not any combination of medium parameters can be found in nature. This aspect is very important and requires further consideration, but is beyond the scope of this paper. Nevertheless, from the obtained dependencies (see Figure 3) one can draw a general conclusion that from two liquids with the same optical (n) and acoustic properties (w,v), the liquid with the larger value of the molecule polarizability α is more preferable. This study reviewed the literature on optical, acoustic and elasto-optic properties, as well as acousto-optic figure of merit, and the collected data for about 100 liquids are presented in Table A1 in Appendix A.

Since the refractive index *n* depends on the wavelength λ, the dispersion of the elasto-optic constant *p* can be estimated using (Equation 21):(28)$$\frac{dp}{d\lambda }\approx \left\{\begin{array}{cc} 2\frac{dn}{d\lambda } & \mathrm{at}\;n-1\ll 1\\ -\frac{4}{n^{3}}\frac{dn}{d\lambda } & \mathrm{at}\;n\gg 2 \end{array}\right..$$

To calculate the dispersion of the elasto-optic constant *p*, the experimental Eykman’s rule (Equation 18) was used, as well as data on the refractive index dispersion n(λ) of both polar and nonpolar liquids. Data for alkanes were related to the range 0.32–0.65 μm at a temperature of 300 K [38], whereas data for alcohols were obtained in the much wider range 0.45–1.55 μm at the same temperature [39]. In addition, data for liquid xenon for the range 0.18–0.65 μm at 162.35 K were taken from [40]. The dynamics of p(λ) depended on the relationship between the refractive index *n* and its optimal value $n_{\mathrm{opt}}$ for maximal *p* (see Table 1): (1) for $n<n_{\mathrm{opt}}$, the elasto-optic constant *p* decreased with the wavelength λ; (2) for $n>n_{\mathrm{opt}}$, the elasto-optic constant increased with λ; and (3) for $n\approx n_{\mathrm{opt}}$, the elasto-optic constant did not depend on λ. The results are shown in Figure 4.

For the crystalline media, the components of the elasto-optic tensor were calculated on the basis of density functional theory to determine the structure of the electronic zone [41,42]. The above method was only implemented numerically, and therefore, there was no analytical formula for the $p_{ij}$ components. It is worth mentioning that a method for estimating the $p_{11}$ and $p_{12}$ components for electro-optic crystals based on electro-optic, optical and acoustic properties was recently proposed [43]. However, this method has not yet found wide application and was only validated for one crystal. At the same time, for liquids, as can be seen from the relationships in Section 2.2, the elasto-optic constant *p* depended only on the refractive index *n*. Therefore, now (1) it is possible to calculate the elasto-optic constant using a simple relationship and (2) the analysis of the results is reduced to the study of a function of one variable. In our opinion, it is one of the fundamental laws of nature that the function p(n) is bounded. However, this law has so far only been shown for liquids in this study.

Let us try to give a physical interpretation of the limitation of the maximum value of the elasto-optic constant *p*. For this purpose, let us use the original relation (Equation 4), which is the definition of this constant, and rewrite it so that it contains relative physical quantities:(29)$$\frac{\Delta n}{n}=\frac{pn^{2}}{2}S.$$

As can be seen from the literature data, for most liquids, 1.3<n<1.6 in the range of transparency (see Figure 1), while the maximum refractive index is n=2.1 for the liquid Se_2_Br_2_ [44] (the experimental value of *p*, however, is unknown). At the same time, the maximum value of the elasto-optical constant is limited to p<0.375 (see Table 1). Finally, we arrive at the following relationship: Δn/n<0.8S, i.e., the relative change in the refractive index is less than the relative strain.

## 6. Conclusions

This article deals with the basics of the theory of the elasto-optic effect in liquids. Starting from first principles, an expression for the acousto-optic figure of merit is derived, which allows one to estimate the optimal ratios of the material parameters of the medium. The physical properties, as well as the calculated acousto-optic figure of merit, of about 100 liquids are summarized. Formulas for estimating the elasto-optic constant from the known refractive index are derived. It was found that the elasto-optic constant of any liquid cannot be greater than 0.375. This fact is of fundamental importance, as it provides a qualitative explanation as to why the elasto-optic constants of known liquid and crystalline media are less than one. In the future, it is planned to develop the proposed method for isotropic dielectrics.

## Figures and Tables

**Figure 1 materials-17-02810-f001:**
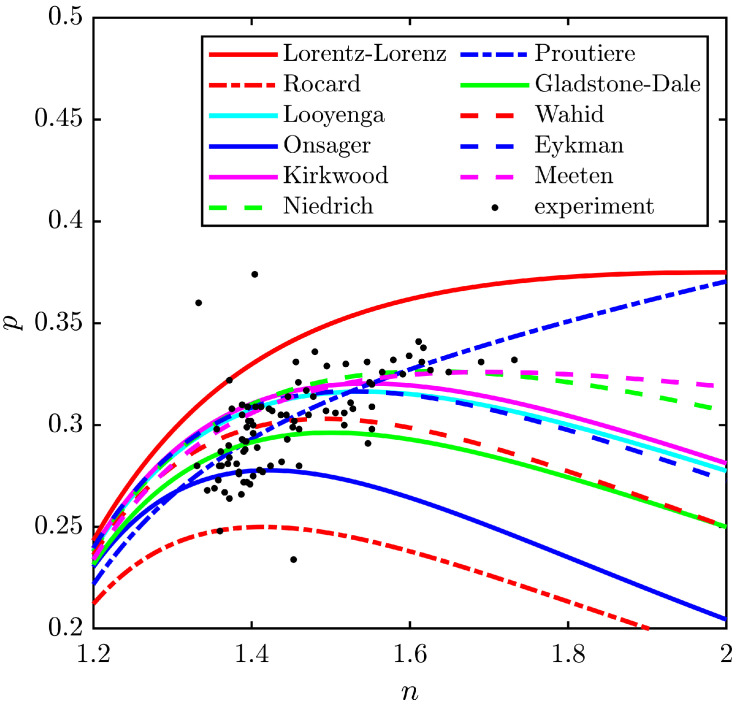
Dependence of the elasto-optic constant of liquids on the refractive index: experimental data and modeling results.

**Figure 2 materials-17-02810-f002:**
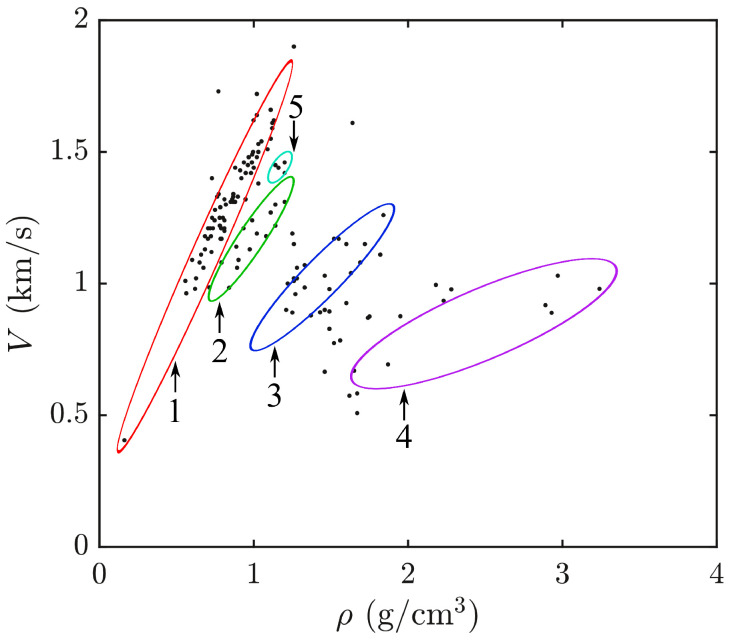
Data on the density and speed of sound for a number of organic liquids: 1—saturated hydrocarbons; 2—replacement of one hydrogen with F or Cl; 3—from two to six hydrogens are replaced with F or Cl; 4—from one to four hydrogens are replaced with Br; 5—aromatic hydrocarbons.

**Figure 3 materials-17-02810-f003:**
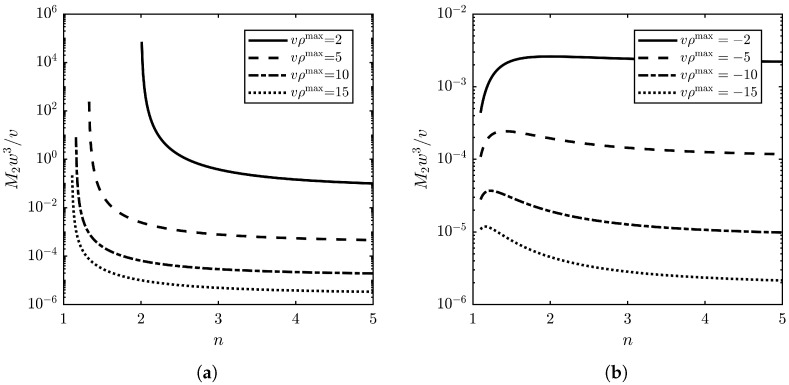
Dependence of the normalized acousto-optic figure of merit of liquid on the refractive index according to the Lorentz–Lorenz model: (**a**) w>0, v>0; (**b**) w<0, v<0.

**Figure 4 materials-17-02810-f004:**
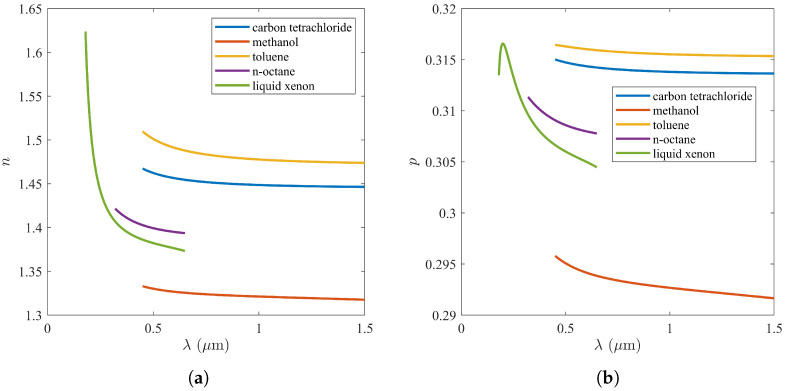
Dispersion curves for some liquids: (**a**) experimental data for refractive index; (**b**) calculated elasto-optic constant.

**Table 1 materials-17-02810-t001:** Optimal refractive index and maximum elasto-optic constant for different models.

	Lorentz	Rocard	Looyenga	Onsager	Kirkwood	Niedrich	Gladstone	Eykman	Wahid	Meeten
$n_{\mathrm{opt}}$	2.000	1.414	1.540	1.421	1.547	1.625	1.500	1.581	1.492	1.696
max(*p*)	0.375	0.250	0.316	0.278	0.320	0.326	0.296	0.316	0.303	0.326

## Data Availability

Data are contained within this article.

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
