# Peer review of "Evaluation of the Acousto-Optic Figure of Merit and the Maximum Value of the Elasto-Optic Constant of Liquids"

_materials, 2024, doi:10.3390/ma17122810_

Round 1

Reviewer 1 Report

Comments and Suggestions for Authors

Comments on the manuscript

The manuscript investigates the elasto-optic properties of liquids, focusing on calculating the elastic-optic constant based only on the refractive index. Experimental data on refractive index dispersion and density functional theory calculations are utilized to derive formulas for estimating the elasto-optic constant. The authors highlighted the importance of the acousto-optic figure of merit in optimizing the selection of interaction mediums for acousto-optic devices. Their findings suggest a maximum value limitation for the last-optic constant of liquids.

In my assessment, the theoretical findings of this work contribute to a comprehensive analysis of the elasto-optic properties of liquids, a topic of considerate interest. Hence, I support the publication of this work, on the condition that certain concerns are addressed. Please find my suggestions and comments for the author below.

1.      The results of one hundred liquid calculations are quite impressive. How feasible is it to utilize elastic-optic modulation for adjusting the high-Q resonances in metasurfaces? [Nano Lett. 2020, 20, 9, 6351–6356].

2.      I wonder if the authors thoroughly discussed the potential limitations or assumptions of the numerical implementation method for calculating the elasto-optic constant, considering the complexities involved in modeling the elasto-optic properties of liquids.

3.      The authors mentioned a maximum value limitation for the elasto-optic constant of liquids. How does this limitation translate to practical applications in acousto-optic devices or other optical systems? Furthermore, how does it influence the design and optimization of such systems? Addressing these practical considerations would enhance the overall usefulness of the research.

Comments on the Quality of English Language

readable

Author Response

Thank you for taking the time to read our article and providing important feedback on it! The answers are in the attachment.

Reviewer 2 Report

Comments and Suggestions for Authors

Comments to Authors

The reviewed paper proposed a method to evaluate the acousto-optic figure of merit of liquids. And the relationship of critical parameter influencing acousto-optic figure of merit is studied theoretically, including density, dielectric permittivity, refractive index, acoustic velocity and elasto-optic constant of liquids. I believe that the topic is interesting and the results are reasonable. However, I think there are some issues that need to be addressed before publication:

1.       The frequency range of acoustic wave, in which the proposed method for evaluation the acousto-optic figure of merit of liquids is applicable should be clarified.

2.       In section 3, the authors reviewed several published permittivity models of liquids.  The assumptions of some of the models were mentioned. However, there is no discussion on the purposed of the made assumption or the applicable range of the reviewed models. In addition to the numerical comparison in section 4, the applicable range of the reviewed models should be major factors to be considered for further selection

3.       In Figure 1, please indicate which data points are from Reference 13 and which ones are from Reference 24.

4.       In Line 124 to Line130, the authors state that Lorentz-Lorentz and Proutiere model established a clear difference form other model. Should we consider the possibility that the errors of these two models are too large within this range?

5.       The authors provide the maximal value of density in Line 153. Then, it is supposed to provide that Figure 3 shows the maximum or minimum value of figure of merit.

Typos are listed but not limited to the following:

1.       Line 205, “this is one of the fundamental laws of nature that that the function”, there should be only one “that”.

Comments on the Quality of English Language

Typos are listed but not limited to the following:

1.       Line 205, “this is one of the fundamental laws of nature that that the function”, there should be only one “that”.

Author Response

(The authors gave the same response as above.)

Round 2

Reviewer 2 Report

Comments and Suggestions for Authors

The authors have answered all of the comments, and made corresponding revisions in the manuscript. My suggestion is that it can be accepted in the current form.